# Stepwise Expansion of Antimicrobial Stewardship Programs and Its Impact on Antibiotic Use and Resistance Rates at a Tertiary Care Hospital in Korea

Dong Hoon Shin,[a] Hyung-Sook Kim,[b] Eunjeong Heo,[b] Myoung-jin Shin,[c] Nak-Hyun Kim,[a] (ORCID) Hyunju Lee,[d] Jeong Su Park,[e] Kyoung Un Park,[e] Jongtak Jung,[a] (ORCID) Kyoung-Ho Song,[a] Minsun Kang,[f] Jaehun Jung,[f,g] (ORCID) Eu Suk Kim,[a] (ORCID) Hong Bin Kim[a]

[a]Department of Internal Medicine, Seoul National University Bundang Hospital, Seoul National University College of Medicine, Gyeonggi-do, Republic of Korea
[b]Department of Pharmacy, Seoul National University Bundang Hospital, Gyeonggi-do, Republic of Korea
[c]Infection Control Center, Seoul National University Bundang Hospital, Gyeonggi-do, Republic of Korea
[d]Department of Pediatrics, Seoul National University Bundang Hospital, Seoul National University College of Medicine, Gyeonggi-do, Republic of Korea
[e]Department of Laboratory Medicine, Seoul National University Bundang Hospital, Seoul National University College of Medicine, Gyeonggi-do, Republic of Korea
[f]Artificial Intelligence and Big-Data Convergence Center, Gil Medical Center, Gachon University College of Medicine, Incheon, Republic of Korea
[g]Department of Preventive Medicine, Gachon University College of Medicine, Incheon, Republic of Korea

**ABSTRACT** To optimize antibiotic use, the US CDC has outlined core elements of antimicrobial stewardship programs (ASP). However, they are difficult to implement in limited-resource settings. We report on the successful implementation of a series of ASP with insufficient number of infectious diseases specialists. We retrospectively collected data regarding antibiotic administration and culture results of all patients admitted to a tertiary care teaching hospital, Seoul National University Bundang Hospital (SNUBH), from January 2010 to December 2019. Trends of antibiotic use and antibiotic resistance rates were compared with those from Korean national data. Trend analyses were performed using nonparametric, two-sided, correlated seasonal Mann–Kendall tests. Total antibiotic agent usage has significantly decreased with ASP implementation at SNUBH since 2010. National claim data from tertiary care hospitals have revealed an increase in the use of all broad-spectrum antibiotics except for third-generation cephalosporins (3GC). In contrast, at SNUBH, glycopeptide and fluoroquinolone use gradually decreased, and 3GC and carbapenem use did not significantly change. Furthermore, the rate of colonization with methicillin-resistant *Staphylococcus aureus* showed a consistently decreasing trend, while that with 3GC- and fluoroquinolone-resistant *Escherichia coli* significantly increased. Unlike the national rate, the rate of colonization with antibiotic resistant-*Klebsiella pneumoniae* did not increase and that of 3GC- and fluoroquinolone-resistant *Pseudomonas aeruginosa* significantly decreased. Stepwise implementation of core ASP elements was effective in reducing antibiotic use despite a lack of sufficient manpower. Long-term multidisciplinary teamwork is necessary for successful and sustainable ASP implementation.

**IMPORTANCE** Antimicrobial stewardship programs aimed to optimize antibiotic use are difficult to implement in limited-resource settings. Our study indicates that stepwise implementation of core antimicrobial stewardship program elements was effective in reducing antibiotic use in a tertiary care hospital despite the lack of sufficient manpower.

**KEYWORDS** antimicrobial stewardship program, core elements, multidisciplinary teamwork, antibiotic use, antibiotic resistance

Address correspondence to Hong Bin Kim, hbkimmd@snu.ac.kr.

The authors declare a conflict of interest. This work was supported by the Seoul National University Bundang Hospital Research Fund (grant number 02-2021-024). The funders had no role in study design, data collection and interpretation, or the decision to submit the work for publication.

Antimicrobial resistance (AMR) has been a global issue for decades. An increase in antibiotic use in the past has exerted selective pressure on susceptible bacteria, leading to the rise of AMR (1). Previous studies have reported that 30 to 50% of

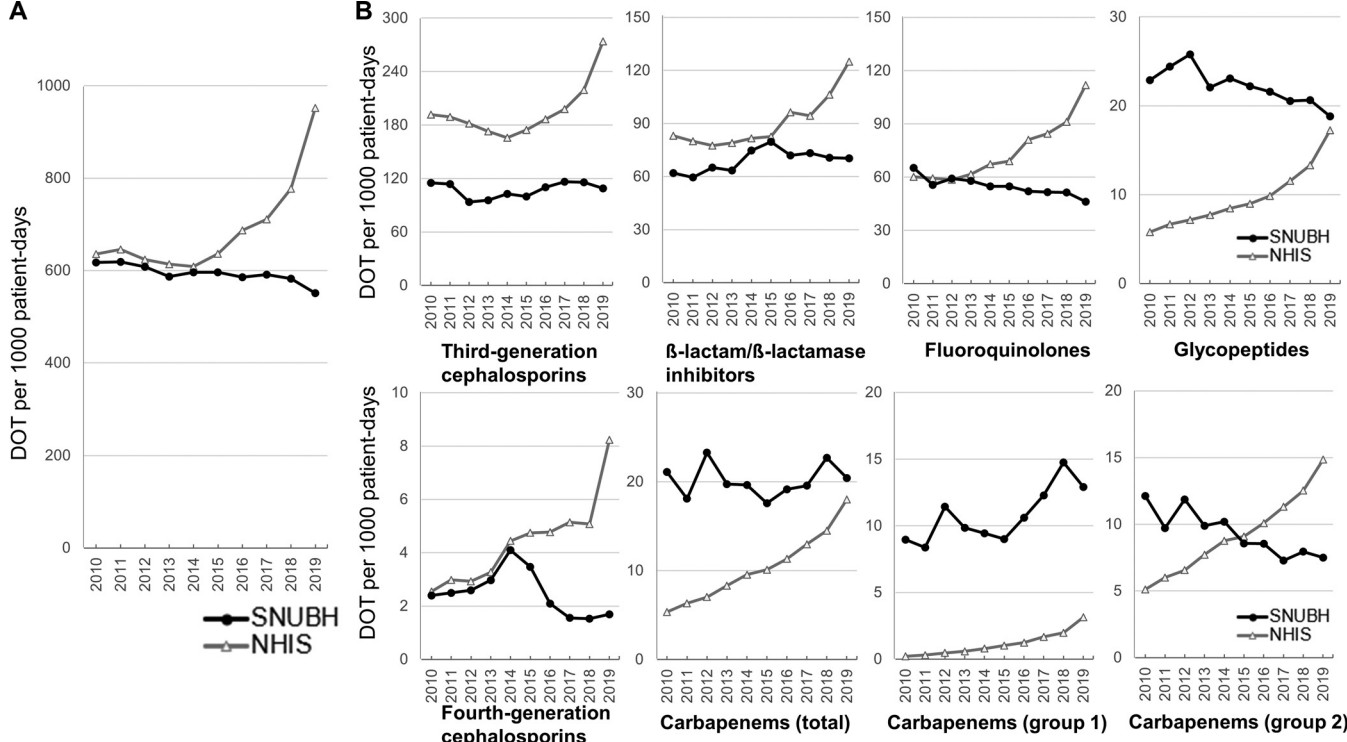

**FIG 1** Trends of antibiotic use (days of therapy [DOT] per 1,000 patient-days) at Seoul National University Bundang Hospital (SNUBH) from 2010 to 2019. (A) Total antibiotic use. (B) Broad-spectrum antibiotic use. NHIS, National Health Insurance Service; Carbapenems (total), all carbapenems; Carbapenems (group 1): ertapenem; Carbapenems (group 2): meropenem, imipenem-cilastatin, and doripenem.

antibiotic prescriptions in hospitals are inappropriate and that broad-spectrum antibiotics are overprescribed (2, 3). Antibiotic misuse has led to increased AMR rates, resulting in significant morbidity and mortality (4). In addition to infection prevention and control, aggressive antimicrobial stewardship programs (ASP) to ensure optimal antibiotic use are urgently needed to reduce AMR rates (5).

To optimize the use of antibiotics, the United States Centers for Disease Control and Prevention (CDC) encouraged all hospitals in the United States to implement ASP and outlined the core elements of hospital-based ASP (6). Some hospitals applied these guidelines and successfully reduced the duration of antibiotic use, the length of hospital stays, and the rate of *Clostridioides difficile* infection (7, 8). However, in many countries, including Korea, it is difficult to apply ASP core elements at a national level due to low compliance by clinicians, a lack of expertise, and the absence of an appropriate reward structure (9).

Here, we summarize the findings from a tertiary care hospital with limited resources where an ASP was successfully implemented for over a decade and analyze the effects of ASP on antibiotic use and AMR rates.

## RESULTS

**Antibiotic use.** Figure 1 shows the days of therapy (DOT) with antibiotics per 1,000 patient-days over the course of 10 years at Seoul National University Bundang Hospital (SNUBH). Total antibiotic use significantly decreased by approximately 10.80% from 2010 to 2019 (Mann–Kendall test, $P < 0.01$; Table S1 in the supplemental material). Glycopeptide ($P < 0.01$) and fluoroquinolone (FQ, $P < 0.01$) use gradually decreased, while third-generation cephalosporin (3GC, $P = 0.48$), fourth-generation cephalosporin ($P = 0.29$), and beta-lactam/beta-lactamase inhibitor ($P = 0.21$) use did not significantly change. There was no significant change in carbapenem use, although use of the group 1 carbapenem ertapenem increased ($P = 0.02$). Total antibiotic use was lower in SNUBH data than in Korean National Health Insurance Service (NHIS) data. In addition,

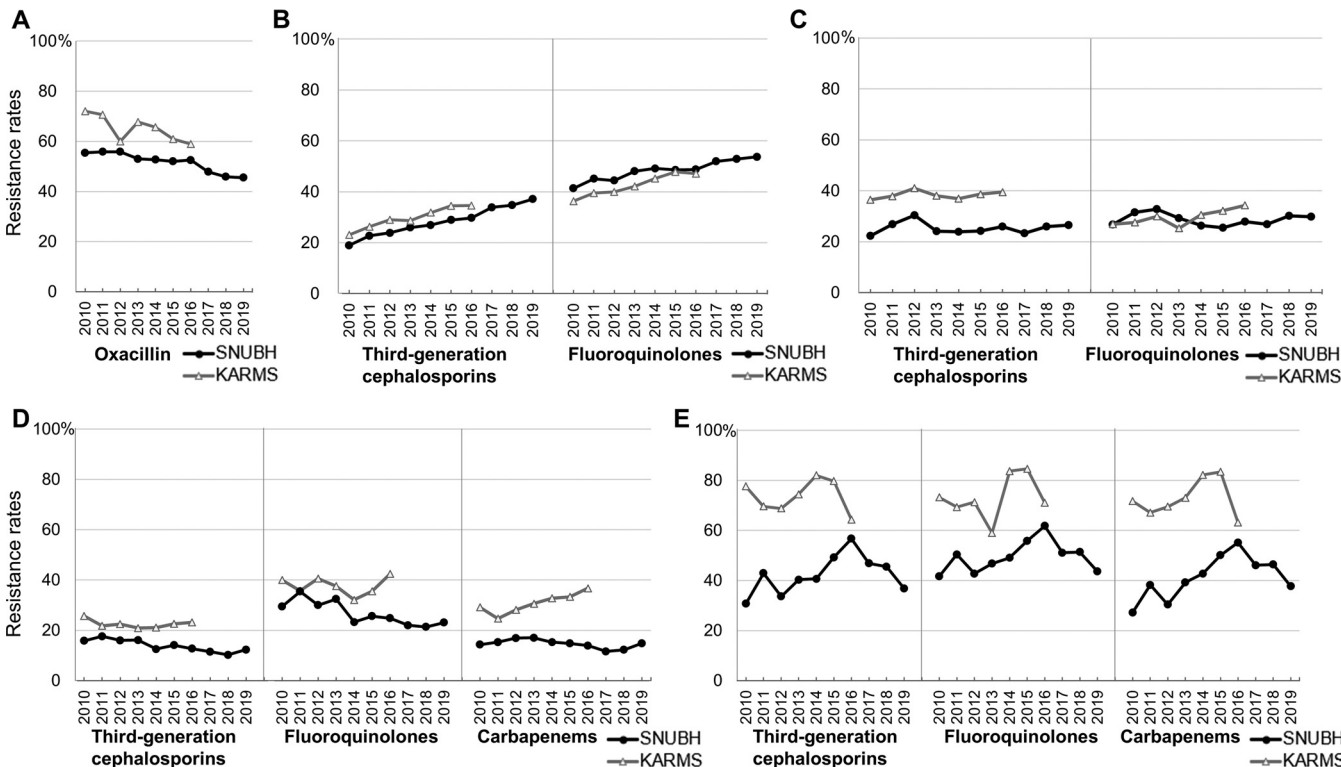

**FIG 2** Changes in antibiotic resistance rates at SNUBH from 2010 to 2019. (A) *Staphylococcus aureus*, (B) *Escherichia coli*, (C) *Klebsiella pneumoniae*, (D) *Pseudomonas aeruginosa*, (E) *Acinetobacter baumannii*. KARMS: Korean Antimicrobial Resistance Monitoring System. Antimicrobial susceptibility was determined using the disk-diffusion method or the Vitek 2 (bioMérieux, Marcy l'Étoil, France). Each isolate was classified as resistant or nonresistant according to Clinical and Laboratory Standards Institute criteria. Proportions of isolates of each bacteria species which were resistant to the following antibiotics were assessed: oxacillin, third-generation cephalosporins (cefotaxime or ceftazidime), fluoroquinolones (ciprofloxacin), and carbapenems (imipenem). The proportions of carbapenem-resistant *E. coli* and *K. pneumoniae* at SNUBH remained less than 1.14%.

the use of every broad-spectrum antibiotic except 3GC continuously increased at other Korean tertiary care hospitals. Although glycopeptide and carbapenem use were higher in SNUBH data than in NHIS data, all types of broad-spectrum antibiotic prescriptions except for group 1 carbapenem showed no increase, and at least maintained a stable level, at SNUBH.

**Antimicrobial resistance rates.** Figure 2 shows the changes in AMR rates by bacterial species. At SNUBH, the methicillin-resistance rate of *Staphylococcus aureus* significantly decreased (Mann–Kendall test, $P < 0.01$; Table S2). The rates of 3GC and FQ resistance increased in *Escherichia coli* ($P < 0.01$), did not significantly change in *Klebsiella pneumoniae* ($P = 1.00$), and decreased in *Pseudomonas aeruginosa* ($P = 0.01$). The proportion of carbapenem-resistant *Acinetobacter baumannii* (CRAB) seemed to increase, but the trend was not statistically significant ($P = 0.07$).

The trend of methicillin-resistant *S. aureus* (MRSA) colonization was consistently lower, and that of 3GC- and FQ-resistant *E. coli* colonization was higher, in the SNUBH data than in the Korean Antimicrobial Resistance Monitoring System (KARMS) data. National data revealed increasing trends for FQ-resistant *K. pneumoniae* and carbapenem-resistant *P. aeruginosa* (CRPA), but SNUBH data showed no significant changes in the resistance rates for these organisms. Although the carbapenem-resistance rate of *A. baumannii* in the SNUBH data was 40% lower than that in the KARMS data in 2010, the difference between the two decreased over time.

## DISCUSSION

The ASP team at SNUBH not only introduced new activities one by one but also ensured that they were continued. With the implementation of strong ASP at SNUBH, not only total antibiotics use but also broad-spectrum antibiotic prescriptions except

for group 1 carbapenem showed no increase compared to nationwide trends. Although AMR rates of *E. coli* increased like that in the nationwide data, the AMR rates of other important pathogens did not increase at SNUBH.

There are a few studies on the application and outcomes of ASP implementation. Although most of them were single-center studies, they showed that the use of antibiotics (especially carbapenems) and the rates of CRPA and CRAB decreased after ASP implementation (10, 11). However, many institutions with insufficient medical resources are finding it difficult to realize the necessity and cost-effectiveness of ASP. Limited human and fiscal resources are regarded as the main barriers to ASP implementation in the Asia-Pacific region (12). Despite such environmental constraints, ASP implementation in hospitals has been found to reduce antibiotic consumption and hospital-acquired infection rates without worsening clinical outcomes (13). While Korea is a relatively high-income country, it has insufficient support for ASP: only one infectious disease (ID) physician per 372 beds was involved in the ASP on a part-time basis, and there are less than 300 ID physicians among the population of 50 million (14, 15). Moreover, each ID physician performs various roles, including diagnosis and treatment of ID, outpatient-based antibiotic therapy administration, infection prevention and control, education, research, planning responses to emerging ID, and ASP. At SNUBH, although the ASP team had only one full-time worker (an ID pharmacist), a low level of antibiotic prescription was maintained. Based on these results, we attempted to formulate an appropriate ASP implementation model with limited resources.

For implementing new ASP activities, interdepartmental teamwork involving pharmacists, laboratory medicine physicians, and infection control nurses, as well as support from the senior leadership, are vital. To increase clinician compliance, we gradually expanded ASP activities based on small successes. For example, the duration of routine perioperative antibiotic prophylaxis was changed to less than 5 days, and this was reflected in clinical decision support systems, as shown in a previous study (16). After checking that the rate of surgical site infections did not increase, this duration was gradually decreased to 3 days, 2 days, and 24 h, with the consent of surgeons. This was made possible by the implementation of a national hospital evaluation program for arthroplasty, gastrectomy, and hysterectomy, which began in 2007 (17). The application of antibiotic prophylaxis guidelines was subsequently extended to other surgeries. Moreover, it was relatively simple and intuitive to reduce redundant combinations of metronidazole or clindamycin with other anti-anaerobic antibiotic agents, and this allowed clinicians to recognize the importance of ASP (18). Based on these successes, we expanded the monitoring of specific antibiotic use, resulting in changes such as modification of the FQ administration route from intravenous to oral and intervention for all long-term (over 2 weeks) antibiotic prescriptions.

The designation of an ID pharmacist as a co-leader was important in maintaining the expanded ASP. The addition of new activities alongside the continuation of existing ones led to a steady decrease in total antibiotic use. Also, in the case of ASP activities in which the ID pharmacist was the lead, physician adherence was maintained over 79% through continuous monitoring of the ASP performance (18, 19). Because only a few pharmacists are available to lead hospital ASP in Korea, there is an urgent need to support human resource development. To further train ID pharmacists and improve their expertise, various certification programs about ASP which include academic societies need to be introduced in Korea.

In this study, the proportion of MRSA decreased, which might have been related to decreased FQ use, as mentioned in a previous study (20). Although the reason for this remains unclear, the trend has been observed worldwide (21). Nevertheless, the use of glycopeptides has not declined globally (22) or in other tertiary care hospitals in Korea. Therefore, the decrease in glycopeptide use at SNUBH was not only due to a decreased MRSA rate but also due to several ASP activities. For example, preauthorization of glycopeptides and the use of routine multiplex PCR in patients with bacteremia with Gram-positive cocci in clusters resulted in reduced durations of vancomycin administration

(23). Thus, regardless of the decreased MRSA rate, aggressive ASP intervention is required to avoid unnecessary glycopeptide use in patients with conditions such as community-acquired pneumonia and neutropenic fever (24, 25).

Beta-lactam/beta-lactamase inhibitors and 3GC were used to treat most of the common infections, such as hepatobiliary and gastrointestinal tract infection, skin and soft tissue infection, urinary tract infection, and pneumonia. Although these antibiotics were not restricted or monitored if used for less than 2 weeks, their use did not significantly increase at SNUBH, possibly because education about the optimal empirical antibiotic therapy for common infections was periodically provided to physicians. Since the ASP team cannot monitor the prescription of each antibiotic, ASP activities with a low work burden, such as education, should be consistently provided.

Although the use of carbapenems, particularly group 2 carbapenems, did not significantly change, ertapenem use increased at SNUBH, possibly due to an increase in resistance to 3GC (especially in *E. coli*). Replacing group 2 carbapenems with ertapenem for empirical antibiotic therapy in critically ill patients might help reduce CRPA and CRAB colonization; however, previous studies on this strategy have obtained conflicting results (26). At SNUBH, the rate of colonization with CRPA did not increase. Furthermore, the absolute rate of CRAB colonization remained lower than those calculated from the data of the KARMS and the Korea Global Antimicrobial Resistance Surveillance System (27). However, since the rate of CRAB colonization increased despite no increase in carbapenem use at SNUBH, additional ASP strategies are needed to reduce CRAB colonization, and further prospective trials on this topic are needed.

This study has some limitations. First, because the study was retrospective and various ASP-related activities were performed together, the specific effect of each activity could not be distinguished. However, these activities can be expected to ultimately be effective in reducing antibiotic use and antibiotic resistance rates based on past experience (28). Second, although we could decrease or at least avoid an increase in the use of broad-spectrum antibiotic agents compared with other tertiary care hospitals, the trends of AMR rates of most pathogens were similar to those calculated from KARMS data. Since AMR rates were likely to be associated with several factors other than antibiotic use, such as infection control practices (29), regional increase in AMR, and agricultural use of antibiotics, a causal relationship between decreased antibiotic use and change in antibiotic resistance rates could not be confirmed. To control these factors simultaneously, it is not enough to implement an ASP at one institution, and a nationwide strategy would be needed to reduce AMR rates. In particular, changing the insurance system to induce fewer antibiotic prescriptions for outpatients, which account for about 80.9% of total antibiotic prescriptions in Korea (30), and introducing a system to monitor and regulate the use of veterinary antibiotics, would be helpful based on previous guidelines (31). Last, KARMS and SNUBH data cannot be directly compared because their surveillance systems are different. However, trends of changes in the proportion of antibiotic-resistant bacteria by year could be compared.

In conclusion, stepwise implementation of the core elements of ASP outlined by the US CDC was effective in preventing increases in antibiotic use despite a lack of sufficient manpower. Therefore, beginning with basic activities, step-by-step expansion, and long-term multidisciplinary teamwork can improve the success and sustainability of an ASP model even in settings with limited resources.

## MATERIALS AND METHODS

**Study population and data collection.** This study was conducted at SNUBH, a tertiary care teaching hospital established in 2003 with approximately 900 beds, which was increased to over 1,300 beds in 2013. We retrospectively collected data regarding antibiotic administration and culture results of all patients hospitalized between January 2010 and December 2019. This study was approved by the Institutional Review Board of SNUBH (no. X-2101-663-902), and the requirement for obtaining written informed consent was waived due to the retrospective nature of the study.

**Hospital leadership and stepwise implementation of ASP at SNUBH.** The ASP implemented at SNUBH are summarized in Table 1 (detailed in Table S3 in the supplemental material). At first, one internist and one pediatrician specializing in ID were involved in developing the ASP in 2003. A team

**TABLE 1** Summarized antimicrobial stewardship activities performed for hospitalized patients at Seoul National University Bundang Hospital until 2020[a]

| ASP core elements | Examples | Initiation date |
|---|---|---|
| Hospital leadership | Two ID physicians at the beginning, now five, involved in ASP activities on a part-time basis[b] | March 2003 |
| | CDSS launched | September 2003 |
| | Pharmacy and therapeutics committee (40) | September 2003 |
| Accountability | Creation of an official ASP team consisting of ID specialists, pharmacists, and microbiology laboratory staff | November 2018 |
| Pharmacy expertise | Designation of one full-time ID pharmacist for ASP activities | May 2019 |
| Action | | |
| Preauthorization | Restricted antibiotic approval only | May 2003 |
| | Post-prescription review and feedback through automatic consultation of ID physicians | August 2011 |
| Prospective audit and feedback | Prevention of redundant combinations of anti-anaerobic antibiotics (18) | July 2013 |
| | Intravenous-to-oral conversion (19) | August 2015 |
| | "Shorter is better" campaign | August 2018 |
| Facility-specific treatment guidelines | Shortening the duration of surgical antibiotic prophylaxis via the clinical pathway | April 2015 |
| Tracking | Antibiotic use and outcome measures | May 2003 |
| Reporting | Regular report on antibiotic resistance rates by newsletter | December 2004 |
| Education | Education programs for physicians and pharmacists | March 2016 |

[a]ASP, antimicrobial stewardship program; ID, infectious disease; CDSS, clinical decision support system.
[b]Part-time basis as opposed to full-time equivalents, with full-time equivalents defined as working 52 h per week for ASP-related activities, according to Korean labor laws.

consisting of ID physicians, pharmacists, infection control nurses, and microbiology laboratory staff was unofficially involved in the ASP on a part-time basis. Although this team was not initially recognized as an official department, all members held meetings every week or every other week, ID physicians and pharmacists met every morning for decision-making regarding the ASP, and they finally became an official team in November 2018. Previously, the ASP team did not have full-time medical personnel, but starting in March 2019, a full-time ID pharmacist was officially appointed as a co-leader.

The first action performed as part of the ASP was the preauthorization of restricted antimicrobial agents. Beginning in August 2011, as the number of ID physicians increased, we implemented a system in which ID physicians were automatically consulted before restricted antibiotics were prescribed. At the same time, when blood culture results indicated bacteremia, ID physicians were automatically consulted and the ASP team recommended the type and duration of antibiotics to be administered (32).

After the establishment of these activities, more aggressive ASP activities were gradually added with the consent of the clinicians. For example, the two ID pharmacists (one full-time and one part-time basis), as ASP co-leaders, performed additional activities to reduce the use of redundant anti-anaerobic antibiotics and intravenous FQ (18, 19). Not only was the status of these ASP activities tracked, but also antibiotic prescriptions, especially those prescribed for more than 2 weeks as part of a "shorter is better" campaign, were monitored daily. The ID pharmacist advised physicians to discontinue administration in cases where it was plausible.

In recent years, with the aim of long-term reduction of antibiotic prescription, ID specialists regularly provide lectures to residents and clinicians about antibiotic use for common infections. Additionally, a booklet on antibiotic dosage according to renal function was created and distributed to clinicians.

**Definitions.** Data regarding antibiotic use were extracted from a clinical data warehouse (33), particularly for the following six classes of broad-spectrum antibiotics that are associated with AMR (34): 3GC and fourth-generation cephalosporins, beta-lactam/beta-lactamase inhibitors, FQ, glycopeptides, and carbapenems. Among cephalosporins, 3GC (i.e. ceftriaxone, cefotaxime, and ceftazidime), and fourth-generation cephalosporins (i.e. cefepime) are at high risk of selection of bacterial resistance, so are designated as one of the key targets of ASP, according to the World Health Organiztion (34). Antibiotic use was quantified as days of therapy per 1,000 patient-days. The antibiotic resistance rates of important nosocomial pathogens, including *S. aureus*, *E. coli*, *K. pneumoniae*, *P. aeruginosa*, and *A. baumannii*, were reviewed using the WHONET program, which collects the types of clinical samples and bacteria, and the AMR information of all culture results (35). To avoid duplication, the first isolates of all clinical samples (e.g., blood, sputum, urine, genital swab, and wound swab) from each patient were included. Antimicrobial susceptibility was determined using the disk-diffusion method or the Vitek 2 (bioMérieux, Marcy l'Étoil, France). Each isolate was classified as resistant or nonresistant according to Clinical and Laboratory Standards Institute (CLSI) criteria (36). The proportion of antibiotic-resistant isolates of each bacteria species was assessed as follows: *S. aureus* (oxacillin), *E. coli* and *K. pneumoniae* (3GC [cefotaxime or ceftazidime] and FQ [ciprofloxacin]), and *P. aeruginosa* and *A. baumannii* (3GC [ceftazidime], FQ, and carbapenems).

**Comparisons with Korean national data.** To compare antibiotics use at SNUBH with that at the national level, we reviewed the NHIS database. The NHIS is the sole health insurance provider in Korea

and is responsible for all hospitalization-related costs. We randomly extracted data regarding 40% of all antibiotic-related claims in patients admitted to Korean tertiary care hospitals from 2010 to 2019. Tertiary care hospitals specialize in high-quality medical treatment for severe diseases, as designated by the Ministry of Health and Welfare, with at least 20 departments and 500 beds (37). Antibiotic use was converted into DOT per 1,000 patient-days using annual inpatient data. Data on AMR rates at Korean general hospitals with over 100 beds from 2010 to 2016 were obtained from the national surveillance program, the KARMS (38). Antibiotic-resistant pathogens isolated from clinical samples at 35 designated general hospitals were investigated for each year, and changes in AMR rates were compared between SNUBH and KARMS data. In 2017, a national surveillance program was reorganized into the Korea Global Antimicrobial Resistance Surveillance System (27).

**Statistical analysis.** The trends of antibiotic use and AMR rates were analyzed using nonparametric two-sided correlated seasonal Mann–Kendall tests, with $P < 0.05$ indicating a statistically significant increase or decrease. The magnitude of change per year was estimated using Sen's method (39). Statistical analyses were performed using R version 4.1.1 (R Core Team, Vienna, Austria) and Python statistical software version 3.7.4 (Python Software Foundation, Wilmington, DE).

## SUPPLEMENTAL MATERIAL

Supplemental material is available online only.
**SUPPLEMENTAL FILE 1**, PDF file, 0.3 MB.

## ACKNOWLEDGMENTS

This work was supported by the Seoul National University Bundang Hospital Research Fund (grant number 02-2021-024). The funders had no role in study design, data collection and interpretation, or the decision to submit the work for publication. We declare no other conflict of interest.

This work was presented as a poster abstract (no. 31) at IDWeek, September 29 to October 3, 2021. We thank Editage (www.editage.co.kr) for English language editing.

Study design: H.B.K. Data collection: D.H.S., H.-S.K., E.H., M.-j.S., N.-H.K., and M.K. Data analysis: D.H.S., E.H., Jo.J., K.-H.S., M.K., Ja.J., E.S.K., and H.B.K. Writing the original draft: D.H.S. Subsequent revisions: D.H.S., H.-S.K., E.H., M.-j.S., N.-H.K., H.L., J.S.P., K.U.P., Jo.J., K.-H.S., M.K., Ja.J., E.S.K., and H.B.K. All authors read and approved the final version of the manuscript.

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
