## [Reviewer comments · Microbiology Spectrum]

Microbiology Spectrum

Stepwise Expansion of Antimicrobial Stewardship Programs and Its Impact on Antibiotic Use and Resistance Rates at a Tertiary Care Hospital in Korea

Dong Hoon Shin, Hyung-sook Kim, Eunjeong Heo, Myoung-jin Shin, Nak-Hyun Kim, Hyunju Lee, Jeong Su Park, Kyoung Un Park, Jongtak Jung, Kyoung-Ho Song, Minsun Kang, Jaehun Jung, Eu Suk Kim, and Hong Bin Kim

Corresponding Author(s): Hong Bin Kim, Seoul National University Bundang Hospital

Review Timeline:

Submission Date:	January 27, 2022
Editorial Decision:	February 23, 2022
Revision Received:	March 10, 2022
Accepted:	April 4, 2022

Editor: Bonnie Prokesch

Reviewer(s): The reviewers have opted to remain anonymous.

Transaction Report:

DOI: <https://doi.org/10.1128/spectrum.00335-22>

February 23, 2022

Dr. Hong Bin Kim
Seoul National University Bundang Hospital
Seongnam
Korea (South), Republic of

Re: Spectrum00335-22 (Stepwise Expansion of Antimicrobial Stewardship Programs and Its Impact on Antibiotic Use and Resistance Rates at a Tertiary Care Hospital in Korea)

Dear Dr. Hong Bin Kim:

Link Not Available

Sincerely,

Bonnie Prokesch

Journals Department
Reviewer comments:

Reviewer #1 (Comments for the Author):

The authors present a historical report of the development of their antimicrobial stewardship program (ASP) over approximately 10 years. They trend their antibiotic usage over time as well as their rates of antimicrobial resistance. The paper is a nice descriptive review of developing an ASP over time in Korea. It would add to the literature regarding how to develop ASPs, however there is nothing truly novel in the paper. There are also several points the authors should address first:

1. They mention implementation of ASP's in limited-resource countries. Does Korea fit this definition? While ASP's may be relatively new there, it is generally regarded as a well resourced country in terms of health care capabilities.

2. In their methods they mention collecting medication prescriptions. It is a minor point but they should clarify if this is administrations or prescriptions, as administrations is considered more accurate for the purposes of measuring usage.
3. Their description of their program's development over time is interesting but largely already addressed with Table 1. They should condense this description to only the most relevant points in their program and otherwise reference the Table.
4. They compare their hospital's usage to national usage. However, they do little to tell the reader how comparable these two comparison groups are. They should at least tell the readers more about their own hospital: types of patients seen, average illness severity etc. If they could compare this to similar data for their national usage, that would be helpful. An alternative would be to compare to only those hospitals nationally that have similar types of patients.
5. They mention how some but not all of their usage by antimicrobial decreased over time. But for those agents that didn't decrease, they did at least show a stable rate of usage compared to increases in the national usage over time. They should point this out more definitively. However they should be careful about claiming causation in this comparison.
6. Several times throughout the paper they attempt to make associations between their usage and rates of antimicrobial resistance (AMR). While ASP's exist ultimately to improve rates of AMR, this can be difficult to show causation with. Over 10 years, numerous changes and events in addition to ASP's have been associated with decreased rates of AMR, including Infection Prevention practices, changes in agricultural use of antibiotics, etc. The authors are not able in any way to say their ASP alone impacted AMR, and they should remove all discussion around AMR rates from the manuscript.
7. The discussion includes a nice description of how to grow an ASP over time when faced with limited staffing and administrative resources.
8. They mention pursuing certification by Society for ID Pharmacists. I assume they have no association with this group but there are other certification programs as well. The authors should try to mention others as well or just speak of certification programs generally.
9. Table 1 is well done in that it categorizes their program development over time based on the CDC core elements. The Table is however large so it may be better served as a supplemental table given its size.

Reviewer #2 (Comments for the Author):

Thank you for the opportunity to review your paper. I greatly enjoyed reading it and commend the ASP on all that they've been able to achieve. Please see below for some suggestions:

Page 6, line 94-96: What is the total physician full time equivalents (FTE) for the 4 internists involved in stewardship? If "unofficially involved" means 0 FTEs, please include how many physician hours per week are dedicated to the ASP and if there is a formal structure for which physician performs which activities at which time or if it's more informal, and the physicians do what they can when they can

Page 7, line 113-115: How many total FTEs did you have for PharmDs in ASP?

Page 8, line 135: Consider including a brief descriptor of what WHONET is.

Page 9, line 150-152: Why did you stop collecting KARMS data at 2016? Also, are all hospitals in Korea required to submit data to KARMS?

Results: It may be helpful to include how often ASP recommendations/ID consultations were accepted by providers at your institution.

Page 12, line 195-197 and page 16, line 281-282: Recommend changing the wording of your conclusion statement, as it is hard to conclude that the ASP prevented an increase in resistance rates vs the nationwide trend. The KARMS data stopped at 2016, resistance rates are influenced by many factors, and E.coli resistance significantly increased at your institution.

Page 13, line 218-220: I appreciate that you highlight your step-wise interventions. It's important to grow as you go and involve other stakeholders outside of ASP to implement long-lasting programs

Discussion: Directly showing decreases in AMR from ASP interventions is difficult due to many contributing factors outside of the inpatient setting or ASP influence. Consider adding more references and discussion about this phenomenon. Also consider discussing what some of those outside factors are in Korea. Is there heavy use of antimicrobials in the outpatient setting that may lead to increased AMR despite your inpatient ASP program?

Staff Comments:

Preparing Revision Guidelines

To submit your modified manuscript, log onto the eJP submission site at <https://spectrum.msubmit.net/cgi-bin/main.plex>. Go to

Author Tasks and click the appropriate manuscript title to begin the revision process. The information that you entered when you first submitted the paper will be displayed. Please update the information as necessary. Here are a few examples of required updates that authors must address:

Please return the manuscript within 60 days; if you cannot complete the modification within this time period, please contact me. If you do not wish to modify the manuscript and prefer to submit it to another journal, please notify me of your decision immediately so that the manuscript may be formally withdrawn from consideration by Microbiology Spectrum.

Thank you for the opportunity to review your paper. I greatly enjoyed reading it and commend the ASP on all that they've been able to achieve. Please see below for some suggestions:

Page 6, line 94-96: What is the total physician full time equivalents (FTE) for the 4 internists involved in stewardship? If "unofficially involved" means 0 FTEs, please include how many physician hours per week are dedicated to the ASP and if there is a formal structure for which physician performs which activities at which time or if it's more informal, and the physicians do what they can when they can

Page 7, line 113-115: How many total FTEs did you have for PharmDs in ASP?

Page 8, line 135: Consider including a brief descriptor of what WHONET is.

Page 9, line 150-152: Why did you stop collecting KARMS data at 2016? Also, are all hospitals in Korea required to submit data to KARMS?

Results: It may be helpful to include how often ASP recommendations/ID consultations were accepted by providers at your institution.

Page 12, line 195-197 and page 16, line 281-282: Recommend changing the wording of your conclusion statement, as it is hard to conclude that the ASP prevented an increase in resistance rates vs the nationwide trend. The KARMS data stopped at 2016, resistance rates are influenced by many factors, and *E. coli* resistance significantly increased at your institution.

Page 13, line 218-220: I appreciate that you highlight your step-wise interventions. It's important to grow as you go and involve other stakeholders outside of ASP to implement long-lasting programs

Discussion: Directly showing decreases in AMR from ASP interventions is difficult due to many contributing factors outside of the inpatient setting or ASP influence. Consider adding more references and discussion about this phenomenon. Also consider discussing what some of those outside factors are in Korea. Is there heavy use of antimicrobials in the outpatient setting that may lead to increased AMR despite your inpatient ASP program?

Reviewer comments:

Reviewer #1 (Comments for the Author):

The authors present a historical report of the development of their antimicrobial stewardship program (ASP) over approximately 10 years. They trend their antibiotic usage over time as well as their rates of antimicrobial resistance. The paper is a nice descriptive review of developing an ASP over time in Korea. It would add to the literature regarding how to develop ASPs, however there is nothing truly novel in the paper. There are also several points the authors should address first:

1. They mention implementation of ASP's in limited-resource countries. Does Korea fit this definition? While ASP's may be relatively new there, it is generally regarded as a well resourced country in terms of health care capabilities.

→ Response

As the reviewer pointed out, in 2019, gross domestic product per capita of Korea was 23rd in the world (*OECD data. <https://data.oecd.org/gdp/gross-domestic-product-gdp.htm>*), and health spending per capita was 23rd in the world (*OECD data. <https://data.oecd.org/healthres/health-spending.htm>*), making it a relatively well-resourced country in terms of health care capabilities. However, considering the scale of health spending, Korea lacks manpower for antimicrobial stewardship programs (ASP), since only 3 medical personnel (median) were involved in ASP, and only 6% of hospitals had full-time workers when surveyed [large hospitals with more than 500 beds (*B Kim et al. 2019. J Hosp Infect*)]. Also, Korean ASP guidelines did not exist until 2020. In a situation where national support was insufficient, ASP activities were settled through the efforts within our hospital

with a few staffs, and this environment was called a limited-resource setting. To avoid misunderstanding, the text has been modified as follows.

In the Abstract section, lines 34-36:

However, they are difficult to implement in limited-resource settings. We report on the successful implementation of a series of ASP with insufficient number of infectious diseases specialists.

In the Discussion section, lines 207-208:

However, many institutions with insufficient medical resources are finding it difficult to realize the necessity and cost-effectiveness of ASP.

In the Discussion section, lines 212-217:

While Korea is a relatively high-income country, there is insufficient support for ASP: only one ID physician per 372 beds was involved in the ASP on a part-time basis, and there were less than 300 ID physicians among the 50 million population (26, 27). Moreover, each ID physician performs various roles, including diagnosis and treatment of ID, outpatient-based antibiotic therapy administration, infection prevention and control, education, research, planning a response to emerging ID, and ASP.

2. In their methods they mention collecting medication prescriptions. It is a minor point but they should clarify if this is administrations or prescriptions, as administrations is considered more accurate for the purposes of measuring usage.

→ Response

Thank you for the valuable suggestion. To clarify the meaning, prescriptions were changed to administrations.

In the Abstract section, lines 38-40:

We retrospectively collected data regarding antibiotic administration and culture results of all patients admitted to a tertiary care teaching hospital (SNUBH) from January 2010 to December 2019.

In the Methods section, lines 91-93:

We retrospectively collected data regarding antibiotic administration and culture results of all patients hospitalized between January 2010 and December 2019.

3. Their description of their program's development over time is interesting but largely already addressed with Table 1. They should condense this description to only the most relevant points in their program and otherwise reference the Table.

→ Response

As per the reviewer's recommendation, we have condensed the description in the Methods section.

In the Methods section, lines 98-102:

The ASP implemented at SNUBH have been summarized in Table 1 (detailed in Supplementary Table 1). At first, one internist and one pediatrician specialized in infectious diseases (ID) were involved in developing the ASP in 2003. A team consisting of ID physicians, pharmacists, infection control nurses, and microbiology laboratory staff was unofficially involved in the ASP on a part-time basis.

In the Methods section, lines 108-111:

The first action performed as a part of the ASP was preauthorization of restricted antimicrobial agents. Since August 2011, as the number of ID physicians increased, we implemented a system in which ID physicians were automatically consulted before restricted antibiotics were prescribed.

In the The first action performed as a part Methods section, lines 117-121:

Not only the status of these ASP activities was tracked, but also antibiotic prescriptions - especially those prescribed more than 2 weeks as part of a "shorter is better" campaign - were monitored daily. The ID pharmacist advised the physicians to discontinue administration in cases where it was plausible.

In the The first action performed as a part Methods section, lines 122-125:

In recent years, the aim of long-term reduction of antibiotic prescription, ID specialists regularly provide lectures to residents and clinicians about antibiotic use for common infections. Additionally, a booklet on antibiotic dosage according to renal function was created and distributed to clinicians.

4. They compare their hospital's usage to national usage. However, they do little to tell the reader how comparable these two comparison groups are. They should at least tell the readers more about their own hospital: types of patients seen, average illness severity etc. If they could compare this to similar data for their national usage, that would be helpful. An alternative would be to compare to only those hospitals nationally that have similar types of patients.

→ **Response**

Thank you for your valuable comment. We reviewed the Korean National Health Insurance Service (NHIS) database for the use of antibiotics at the tertiary care hospitals. Tertiary care hospitals in Korea are designated by the Ministry of Health and Welfare every 3 years, and are defined as hospitals that specialize in high-quality medical treatment for severe diseases with at least 20 departments and 500 beds. In addition, the condition must be satisfied that over 30% of hospitalized patients are diagnosed with about 480 designated specialized diseases, and less than 14% of that are diagnosed with about 600 non-specialized diseases (*S Kwon et al. 2015. Republic of Korea Health System Review. Vol.5 No.4. Manila: World Health Organization*). As of 2014, the mean number of beds in tertiary care hospitals was

1029 (standard deviation 405), and our hospital ranked 10th among 43 tertiary care hospitals. Although it was not possible to compare average illness severity with other tertiary care hospitals, it was indirectly identified that they have similar types of patients by checking the size of the hospitals. This has been mentioned as follows:

In the Methods section, lines 148-152:

We randomly extracted data regarding 40% of all antibiotic-related claims in patients admitted to Korean tertiary care hospitals from 2010 to 2019. Tertiary care hospitals are specialized in high-quality medical treatment for severe diseases designated by the Ministry of Health and Welfare with at least 20 departments and 500 beds (18).

5. They mention how some but not all of their usage by antimicrobial decreased over time. But for those agents that didn't decrease, they did at least show a stable rate of usage compared to increases in the national usage over time. They should point this out more definitively. However, they should be careful about claiming causation in this comparison.

→ Response

We appreciate your recommendation. To emphasize that the use of broad-spectrum antibiotics did not increase, the following modifications were made.

In the Results section, lines 179-181:

Although glycopeptide and carbapenem use were higher in SNUBH data than in NHIS data, all kinds of broad-spectrum antibiotic prescriptions except group 1 carbapenem showed no increase, and at least maintained a stable level at SNUBH.

In the Discussion section, lines 199-202:

The ASP team at SNUBH not only introduced new activities one by one but also ensured that they were continued. With the implementation of strong ASP at SNUBH, not only total antibiotics but also broad-spectrum antibiotics prescriptions except group 1 carbapenem showed no increase as compared to the nationwide trend.

6. Several times throughout the paper they attempt to make associations between their usage and rates of antimicrobial resistance (AMR). While ASP's exist ultimately to improve rates of AMR, this can be difficult to show causation with. Over 10 years, numerous changes and events in addition to ASP's have been associated with decreased rates of AMR, including Infection Prevention practices, changes in agricultural use of antibiotics, etc. The authors are not able in any way to say their ASP alone impacted AMR, and they should remove all discussion around AMR rates from the manuscript.

→ Response

As the reviewer mentioned, it is difficult to confirm a causal relationship between strong ASP and AMR rates based on our study because of external factors such as infection control practices, and changes in AMR rates nationwide. The parts where causation was mentioned in the discussion were revised.

In the Abstract section, lines 54-55:

Stepwise implementation of core ASP elements was effective in reducing antibiotic use despite a lack of sufficient manpower.

In the Discussion section, lines 202-203:

Although AMR rates of *E. coli* increased like nationwide data, that of other important pathogens did not increase at SNUBH.

In the Discussion section, lines 278-289:

Second, although we could decrease or at least avoid an increase in the use of broad-spectrum antibiotic agents compared with other tertiary care hospitals, the trends of AMR rates of most pathogens were similar to those calculated from KARMS data. Since AMR rates were likely to be associated with several factors other than antibiotic use, such as infection control practices (38), regional increase in AMR, and agricultural use of antibiotics, a causal relationship between decreased antibiotic use and change in antibiotic resistance rates could not be confirmed. In order to control these factors simultaneously, it is not enough to implement an ASP at one institution, and a nationwide strategy would be needed to reduce AMR rates. In particular, changing the insurance system to induce fewer antibiotic prescriptions for outpatients that account for about 80.9% of the total antibiotic prescription in Korea (39), and introducing a system to monitor and regulate the use of veterinary antibiotics would be helpful based on previous guidelines (40).

In the Discussion section, lines 293-295:

In conclusion, stepwise implementation of the core elements of ASP outlined by the United States Centers for Disease Control and Prevention was effective in preventing an increase in the use of antibiotics despite a lack of sufficient manpower.

7. The discussion includes a nice description of how to grow an ASP over time when faced with limited staffing and administrative resources.

→ **Response**

We are grateful for the reviewer's comment. Through this study, we tried to emphasize the importance of ASP, and suggest a methodology for ASP implementation in institutions without sufficient manpower.

8. They mention pursuing certification by Society for ID Pharmacists. I assume they have no association with this group but there are other certification programs as well. The authors should try to mention others as well or just speak of certification programs generally.

→ **Response**

Thank you for the excellent suggestion. Although the authors do not have any conflicts of interest with the Society for Infectious Disease Pharmacists, to avoid misunderstandings advocating for specific certification qualifications, the discussion section has been modified as follows:

In the Discussion section, lines 243-245:

To further train ID pharmacists and improve their expertise, various certification programs about ASP including academic societies need to be introduced in Korea.

9. Table 1 is well done in that it categorizes their program development over time based on the CDC core elements. The Table is however large so it may be better served as a supplemental table given its size.

→ **Response**

According to the reviewer's comment, the contents to be emphasized are summarized in Table 1, and detailed contents are described in Supplementary Table 1.

In the Methods section, lines 98-99:

The ASP implemented at SNUBH have been summarized in Table 1 (detailed in Supplementary Table 1).

In Table 1:

Table 1. Summarized antimicrobial stewardship activities performed for hospitalized patients at Seoul National University Bundang Hospital until 2020

ASP core elements	Examples	Initiation
Hospital leadership	• Two ID physicians at the beginning, now five involved in ASP activities on a part-time basis ^a	March 2003

	• CDSS launched	September 2003
	• Pharmacy & therapeutics committee (10)	September 2003
Accountability	• Creation of an official ASP team consisting of ID specialists, pharmacists, and microbiology laboratory staff	November 2018
Pharmacy expertise	• Designation of one full-time ID pharmacist for ASP activities	May 2019
Action	Preauthorization	
	• Restricted antibiotic approval only • Post-prescription review and feedback through automatic consultation of ID physicians	May 2003 August 2011
	Prospective audit and feedback	
	• Prevention of redundant combinations of anti-anaerobic antibiotics (11) • Intravenous to oral conversion (12) • “Shorter is better” campaign	July 2013 August 2015 August 2018
	Facility-specific treatment guidelines	
	• Shortening the duration of surgical antibiotic prophylaxis via the clinical pathway	April 2015
Tracking	• Antibiotic use and outcome measures	May 2003
Reporting	• Regular report on antibiotic resistance rates by newsletter	December 2004
Education	• Education programs for physicians and pharmacists	March 2016

ASP: antimicrobial stewardship program; ID: infectious disease; CDSS: Clinical decision support system

^a Part-time basis is a concept as opposed to full-time equivalents, and full-time equivalents are defined as working 52 hours per week for ASP-related activities according to the labor laws in Korea.

In the Supplementary Table 1:

Supplementary Table 1. Antimicrobial stewardship activities performed for hospitalized patients at Seoul National University Bundang Hospital until 2020

ASP core elements	Examples	Initiation
Hospital leadership	Staff involved in ASP activities (on a part-time basis) ^a at a hospital with over 1300 beds	
	• Two ID physicians (1 internist and 1 pediatrician)	May 2003
	• Three ID physicians (2 internists and 1 pediatrician)	March 2009
	• Four ID physicians (3 internists and 1 pediatrician)	March 2011
	• Five ID physicians (4 internists and 1 pediatrician)	March 2020
	CDSS launched	September 2003 Last updated in 2017
	Pharmacy & therapeutics committee	
	• Establishment of a subcommittee for antibiotics	September 2003
	• Promotion to ASP committee	November 2018
	• Establishment of a new subcommittee for therapeutic drug monitoring of antibiotics	September 2019
Accountability	• Creation of an ASP team consisting of ID specialists, pharmacists, and microbiology laboratory staff	November 2018
	• Regular handshake stewardship for the hematology unit	March 2006
Pharmacy expertise	• ID training for the pharmacists	March 2013
	• Designation of one full-time ID pharmacist for ASP activities	May 2019
Action	Preauthorization	
	• Restricted antibiotic approval only, such as broad-spectrum antibiotics (e.g. carbapenems, glycopeptides, polymyxins, and fourth-generation cephalosporins), antifungals (e.g. newer azoles, liposomal amphotericin-B, and echinocandins), and antivirals (e.g. ganciclovir)	May 2003
	• Post-prescription review and feedback through automatic consultation of ID physicians. If the prescription was approved by an ID physician, antibiotics could be administered for 7 days, and in the absence of a decision, antibiotics could be prescribed for 3 days for urgent use. If the ASP team determined that the antibiotic use was inappropriate, they recommended an alternative treatment in their consultation note, and the pharmacy did not prepare the prescribed antibiotics.	August 2011
	Prospective audit and feedback	
	• Electronic alerts with automatic consultation of ID	August 2011

physicians for patients with positive blood cultures

- Prevention of redundant combinations of metronidazole or clindamycin with other anti-anaerobic antibiotics July 2013
- Prospective review of nine antibiotics: azithromycin, cefoxitin, clindamycin, colistimethate, sulfamethoxazole/trimethoprim, anidulafungin, fluconazole, voriconazole, and acyclovir July 2014
- Intravenous to oral conversion of administration of fluoroquinolone and metronidazole August 2015
 - expanded to six more antibiotics August 2019
- “Shorter is better” campaign targeting antibiotics prescribed for more than 2 weeks. All in-hospital long-term antibiotic prescriptions were monitored daily, and the ID pharmacist advised the physicians to discontinue administration in cases where it was plausible. August 2018

Facility-specific treatment guidelines

- Shortening of duration of antibiotic administration for surgical antibiotic prophylaxis via the clinical pathway
 - less than 5 days June 2003
 - less than 3 days November 2009
 - less than 2 days April 2015
 - less than 24 hours April 2020
- Biannual updating of facility-specific guidelines for solid organ transplantation in collaboration with the transplantation teams April 2009
Last updated in May 2019

Pharmacologic intervention

- Vancomycin loading using the computerized CDSS July 2016
- Renal dosing guidance
 - computerized dosing recommendations April 2006
 - dosing pamphlets released to the prescribers July 2020
- Daily alerts, particularly for the pharmacists, using an ASP review sheet in the electronic medical record November 2016
 - intravenous to oral conversion
 - inappropriate dosing according to indications and renal function
 - drug interactions and adverse events

Rapid diagnostics

	 • Multiplex polymerase chain reaction performed in patients with bacteremia with clusters of gram-positive cocci 	February 2012
Tracking	Antibiotic use measures	
	 • Monitoring surgical antibiotic prophylaxis 	October 2007
	 • Monitoring antibiotic administration data from the clinical database (defined daily dose, day of therapy, and length of therapy per 1000 patient-days) 	July 2014
	 • Hospital-wide point prevalence survey on the appropriateness of antibiotic prescription 	August 2018
	Outcome measures	
	 • Biweekly meetings with the staff at the microbiology laboratory and infection control office  - changed to weekly meetings 	May 2003 March 2017
	 • Daily morning conference with the pharmacists 	March 2013
Reporting	 • Regular report on antibiotic resistance rates for important clinical isolates by newsletter 	December 2004
	 • Regular report on antibiotic use 	April 2009
	 • Regular report on the proportion of clinical consultations for therapeutic drug monitoring of antibiotics administered for over 7 days 	December 2019
Education	 • Education programs for physicians and pharmacists about antibiotic choice, dosage, and treatment duration for common infections such as pneumonia, urinary tract infection, and skin and soft tissue infection 	March 2016
	 • Educational materials developed by elective course internal medicine residents and shared with other physicians 	March 2016
	 • Annual ASP symposium for health-system pharmacists 	October 2019

ASP: antimicrobial stewardship program; ID: infectious disease; CDSS: Clinical decision support system

^a Part-time basis is a concept as opposed to full-time equivalents, and full-time equivalents are defined as working 52 hours per week for ASP-related activities according to the labor laws in Korea.

Reviewer #2 (Comments for the Author):

Thank you for the opportunity to review your paper. I greatly enjoyed reading it and commend the ASP on all that they've been able to achieve. Please see below for some suggestions:

Page 6, line 94-96: What is the total physician full time equivalents (FTE) for the 4 internists involved in stewardship? If "unofficially involved" means 0 FTEs, please include how many physician hours per week are dedicated to the ASP and if there is a formal structure for which physician performs which activities at which time or if it's more informal, and the physicians do what they can when they can.

→ Response

Full time equivalents (FTEs) are defined as working 52 hours per week for antimicrobial stewardship programs (ASP)-related activities according to the labor laws in Korea (*SY Park et al. 2020. Infect Control Hosp Epidemiol*). There is a lack of infectious disease (ID) physicians even in tertiary care hospitals which specializes in high-quality medical treatment for severe diseases with at least 20 departments and 500 beds; there is median one ID physician per 377 beds (*Y Jang et al. 2020. J Korean Med Sci*). Thus, as mentioned in the discussion, each ID physician is involved in various work, including diagnosis and treatment of ID, outpatient-based antibiotic therapy administration, infection prevention and control, education, research, planning a response to emerging ID, and ASP without the formal structure of the ASP team. Since there was no full time designation for ASP among the four ID physicians, the exact time of contribution for ASP by each ID physician was unknown.

The term “unofficially involved” was used as there was no official department in charge of ASP within our hospital until November 2018. However, even before the formal ASP team was formed, the ASP activities listed in Supplementary Table 1 were performed through daily morning conferences with physicians and pharmacists, and biweekly meetings with the microbiology laboratory and infection control office. There were no full-time involved personnel in the ASP team until one full-time ID pharmacist was officially appointed in May 2019. To clarify the meaning, the methods section has been modified as follows:

In the Methods section, lines 102-107:

Although the team was not recognized as an official department initially, they held meetings every or every other week with all members, and ID physicians and pharmacists met every morning for decision making about the ASP, and finally they became an official team in November 2018. Previously, the ASP team did not have full-time medical personnel, but from March 2019, a full-time ID pharmacist was officially appointed as a co-leader.

In the footnote of Supplementary Table 1:

^a Part-time basis is a concept as opposed to full-time equivalents, and full-time equivalents are defined as working 52 hours per week for ASP-related activities according to the labor laws in Korea.

Page 7, line 113-115: How many total FTEs did you have for PharmDs in ASP?

→ Response

Our ASP team consisted of two pharmacists at the beginning, each with 0.4 FTE (clinical pharmacist) and 0.5 FTE (pharmacy resident) in infectious diseases (no FTE). After that, in May 2019, the time for ASP activities of clinical pharmacists increased to 1 FTE. Until today, 1 FTE clinical pharmacist and 0.5 FTE pharmacy resident are dedicated to ASP in our team. The part the reviewer mentioned has been revised.

In the Methods section, lines 115-117:

For example, the two ID pharmacists (one full-time and one part-time basis), as ASP co-leaders, performed additional activities to reduce the use of redundant anti-anaerobic antibiotics and intravenous fluoroquinolones (FQ) (11, 12).

Page 8, line 135: Consider including a brief descriptor of what WHONET is.

→ **Response**

We appreciate your valuable comment. The WHONET program codes the data from all in-hospital laboratories automatically (*JM Stelling et al. 1997. Clin Infect Dis*). The program collects the data of all culture results that are reported in the electronic medical records: the types of clinical samples (e.g. blood, sputum, urine, genital swab, and wound swab), the types of bacteria, and resistance information for each antibiotic at once, except for patient personal information. Thus, the program makes it convenient to collect the in-hospital antimicrobial resistance data. We have added a description as follows:

In the Methods section, lines 132-136:

The antibiotic resistance rate of important nosocomial pathogens, including *Staphylococcus aureus*, *Escherichia coli*, *Klebsiella pneumoniae*, *Pseudomonas aeruginosa*, and *Acinetobacter baumannii*, were reviewed using the WHONET program which collects the types of clinical samples and bacteria, and AMR information of all culture results (16).

Page 9, line 150-152: Why did you stop collecting KARMS data at 2016? Also, are all hospitals in Korea required to submit data to KARMS?

→ **Response**

As the reviewer mentioned, the Korean Antimicrobial Resistance Monitoring System (KARMS) was available until 2016, and after 2017, it was reorganized into a new surveillance system called the Korea Global Antimicrobial Resistance Surveillance System (Kor-GLASS; *H Lee et al. 2018. Euro Surveill*), according to the GLASS of World Health Organization. Kor-GLASS was briefly mentioned in the Discussion section, but it could not be treated as the same nationwide data because the surveillance method was different from KARMS.

KARMS was divided into general hospitals, clinics, and nursing hospitals, and collected resistance rate (%) data of each representative institution (*D Kim, et al. 2017. Ann Lab Med*). Although it was different for each bacteria, most of the culture results were from blood and urine specimens, and the AMR rate for each type of bacteria was calculated by averaging the data of representative institutions. In case of duplication, the first isolate of clinical samples from each patient was included. In this study, we reviewed KARMS data from 2010 to 2016 of general hospitals of over 100 beds, which included a total of 35 hospitals. We have added extra explanation as follows:

In the Methods section, lines 153-159:

Data on the AMR rate at Korean general hospitals with over 100 beds from 2010 to 2016 were obtained from the national surveillance program, the Korean Antimicrobial Resistance Monitoring System (KARMS) (19). Antibiotic-resistant pathogens isolated from clinical samples at 35 designated general hospitals were investigated for each year, and the changes in AMR rate were compared between SNUBH and KARMS data. Since 2017, a national surveillance program was reorganized into Korea Global Antimicrobial Resistance Surveillance System (20).

In the Discussion section, lines 270-272:

Furthermore, the absolute rate of CRAB colonization remained lower than those calculated from the data of the KARMS and Korea Global Antimicrobial Resistance Surveillance System (20).

Results: It may be helpful to include how often ASP recommendations/ID consultations were accepted by providers at your institution.

→ Response

Thank you for the valuable suggestion. Although there was no adherence data for all of the ASP team's recommendations, physicians' adherence for some ASP activities were monitored. ID pharmacists participated as co-leaders in the following ASP activities, and high physician compliance could be maintained: prevention of redundant combinations of anti-anaerobic

antibiotics (monitored for 18 months, the adherence rate 93.9%; *M Kim et al. 2020. Eur J Clin Microbiol Infect Dis*), intravenous to oral conversion of fluoroquinolones (monitored for 4 months, the adherence rate 79.8%; *SM Park et al. 2017. Infect Chemother*), and “Shorter is better” campaign targeting antibiotics prescribed for more than 2 weeks (the adherence rate 95%; data not shown). Although it was not possible to describe a uniform adherence rate for ASP activities, the following part was added to emphasize the ID pharmacist's role in monitoring the performance of ASP and contributing to high adherence.

In the Discussion section, lines 238-241:

The addition of new activities alongside the continuation of existing ASP activities led to a steady decrease in total antibiotic use. Also, in the case of ASP activities in which the ID pharmacist was the lead, the physician's adherence was maintained high over 79% through continuous monitoring of the performance of ASP (11, 12).

Page 12, line 195-197 and page 16, line 281-282: Recommend changing the wording of your conclusion statement, as it is hard to conclude that the ASP prevented an increase in resistance rates vs the nationwide trend. The KARMS data stopped at 2016, resistance rates are influenced by many factors, and E. coli resistance significantly increased at your institution.

→ **Response**

As per the reviewer's valuable comment, it is difficult to confirm a causal relationship between strong ASP and AMR rates based on our study because of external factors such as

infection control practices, and agricultural use of antibiotics. The parts where causation was mentioned in the discussion were revised.

In the Abstract section, lines 54-56:

Stepwise implementation of core ASP elements was effective in reducing antibiotic use despite a lack of sufficient manpower. Long-term multi-disciplinary teamwork is necessary for successful and sustainable ASP implementation.

In the Discussion section, lines 202-203:

Although AMR rates of *E. coli* increased like nationwide data, that of other important pathogens did not increase at SNUBH.

In the Discussion section, lines 278-289:

Second, although we could decrease or at least avoid an increase in the use of broad-spectrum antibiotic agents compared with other tertiary care hospitals, the trends of AMR rates of most pathogens were similar to those calculated from KARMS data. Since AMR rates were likely to be associated with several factors other than antibiotic use, such as infection control practices (38), regional increase in AMR, and agricultural use of antibiotics, a causal relationship between decreased antibiotic use and change in antibiotic resistance rates could not be confirmed. In order to control these factors simultaneously, it is not enough to implement an ASP at one institution, and a nationwide strategy would be needed to reduce AMR rates. In particular, changing the insurance system to induce fewer antibiotic

prescriptions for outpatients that account for about 80.9% of the total antibiotic prescription in Korea (39), and introducing a system to monitor and regulate the use of veterinary antibiotics would be helpful based on previous guidelines (40).

In the Discussion section, lines 293-295:

In conclusion, stepwise implementation of the core elements of ASP outlined by the United States Centers for Disease Control and Prevention was effective in preventing an increase in the use of antibiotics despite a lack of sufficient manpower.

Page 13, line 218-220: I appreciate that you highlight your step-wise interventions. It's important to grow as you go and involve other stakeholders outside of ASP to implement long-lasting programs.

→ Response

We are grateful for your comment. Through this study, we tried to suggest a methodology for ASP implementation in institutions without sufficient manpower. An important aspect we wanted to emphasize in the ASP implementation was the multidisciplinary teamwork of the ASP team, which consisted of ID physicians, pharmacists, infection control nurses, and microbiology laboratory staff. Moreover, by tracking the outcome of ASP activities, the team tried to persuade other clinicians to agree on the necessity of ASP and to get them involved into it. For the ASP team not to get tired, and in attempts to convince other clinicians, we set it up step by step from low hanging fruits (relatively easy activities).

Discussion: Directly showing decreases in AMR from ASP interventions is difficult due to many contributing factors outside of the inpatient setting or ASP influence. Consider adding more references and discussion about this phenomenon. Also consider discussing what some of those outside factors are in Korea. Is there heavy use of antimicrobials in the outpatient setting that may lead to increased AMR despite your inpatient ASP program?

→ Response

We appreciate your recommendation. The increase in antibiotic use and the development of antibiotic resistance were associated (*M Kolář et al. 2001. Int J Antimicrob Agents*), and strong ASP could prevent the increase in antibiotic use. However, according to the reviewer's comment, since there are some contributing factors outside of the inpatient setting, ASP did not directly connect to a decrease in the AMR rates of our hospital.

First, AMR rates of our hospital and KARMS cannot be compared equally as mentioned in the limitations, but it was confirmed that most AMR rates of major pathogens show similar trends (Fig. 2.). AMR rates that increased in KARMS but maintained a similar level in our hospital were fluoroquinolone-resistant *K. pneumoniae*, carbapenem-resistant *P. aeruginosa*, and *A. baumannii*. ASP inside our hospital alone cannot block resistant bacteria introduced from the outside, and infection control practices that prevent horizontal transmission are thought to contribute to lowering the AMR rates (*C-H Chen et al. 2015. Int J Environ Res Public Health*). The non-increasing trends of carbapenem-resistant *P. aeruginosa*, and *A. baumannii* may have occurred because the infection control office strongly recommended for routine culture screening, hand hygiene, and contact precautions in our hospital's intensive care units for those bacteria. Although these efforts are important, it is believed that an

increase in AMR rates in the surrounding area, which is an external factor, has a strong correlation with the increase in the hospital.

Second, antibiotic use in the outpatient setting is also a very important outside factor in terms of nationwide total antibiotic use. Antibiotic use for outpatient accounted for about 80% of total antibiotic use based on the defined daily dose using claim data in Korea (*YK Yoon et al. 2015. Medicine*). However, from 2008 to 2012, total antibiotic use among Korean outpatients did not increase. In the case of urinary tract infections (*B Kim et al. 2019. BMC Infect Dis*), and pediatric infection (*YJ Choe et al. 2019. Korean J Pediatr*), which are common indications for antibiotic prescription in outpatient settings, an increase in total antibiotic use was not confirmed, but an increase in the proportion of broad-spectrum antibiotics such as third generation cephalosporins and amoxicillin-clavulanate was confirmed. Even if the amount of total antibiotic use is maintained at a certain level, since the amount of antibiotics used for outpatients is overwhelmingly larger than inpatients, and broad-spectrum antibiotics are being prescribed more and more, it is necessary to continuously educate clinic physicians about appropriate indication and improve the insurance system to control indiscriminate antibiotic prescription.

Last, agriculture use of antibiotics may also contribute to increased antibiotic resistance (*LM Durso et al. 2014. Curr Opin Microbiol*). In fact, there are reports that some antibiotics have been detected in soil and water (*YS Ok et al. 2010. Environ Monit Assess*), and antibiotic-resistance genes have also been detected depending on the region and season in Korea (*D Son et al. 2018. J Environ Quality*). The problem is that the prescription of veterinary antibiotics has not been monitored, and related previous studies have been limited to changes in antibiotic-resistance genes detected by regions in Korea. Therefore, first, there should be a

social consensus on the appropriate indication of veterinary antibiotics use, and nationwide policies to monitor and regulate the prescription of the antibiotics is needed.

Even though considering both ASP and external factors associated with antibiotic resistance, it is expected that a decrease in the overall antibiotic usage will eventually lead to a decrease in antibiotic resistance (*MS Morehead et al. 2018. Prim Care Clin Office Pract*). The role of the government is important in developing and integrating policies as “One Health” perspective, and the results of our study will serve as a good model for ASP implementation in hospitals across the country, thereby contributing to the policies. The limitations section of the discussion was further reinforced for the part the reviewer commented.

In the Discussion section, lines 278-289:

Second, although we could decrease or at least avoid an increase in the use of broad-spectrum antibiotic agents compared with other tertiary care hospitals, the trends of AMR rates of most pathogens were similar to those calculated from KARMS data. Since AMR rates were likely to be associated with several factors other than antibiotic use, such as infection control practices (38), regional increase in AMR, and agricultural use of antibiotics, a causal relationship between decreased antibiotic use and change in antibiotic resistance rates could not be confirmed. In order to control these factors simultaneously, it is not enough to implement an ASP at one institution, and a nationwide strategy would be needed to reduce AMR rates. In particular, changing the insurance system to induce fewer antibiotic prescriptions for outpatients that account for about 80.9% of the total antibiotic prescription in Korea (39), and introducing a system to monitor and regulate the use of veterinary antibiotics would be helpful based on previous guidelines (40).

April 4, 2022

Prof. Hong Bin Kim
Seoul National University Bundang Hospital
Internal Medicine
166 Gumi-ro, Bundang-gu
Seongnam, Gyeonggi 463-707
Korea (South), Republic of

Re: Spectrum00335-22R1 (Stepwise Expansion of Antimicrobial Stewardship Programs and Its Impact on Antibiotic Use and Resistance Rates at a Tertiary Care Hospital in Korea)

Dear Prof. Hong Bin Kim:

Your manuscript has been accepted, and I am forwarding it to the ASM Journals Department for publication. You will be notified when your proofs are ready to be viewed.

Sincerely,

Bonnie Prokesch
Editor, Microbiology Spectrum

Journals Department
Supplemental Tables: Accept